# The PlayerScore: A Systematic Game Observation Tool to Determine Individual Player Performance in Team Handball Competition

Herbert Wagner [1,2,*], Matthias Hinz [2] , Kevin Melcher [2], Vanja Radic [2] and Jörn Uhrmeister [3]

[1] Department of Sport and Exercise Science, University of Salzburg, Schlossallee 49, 5400 Hallein-Rif, Austria
[2] Department of Sport Science, Otto von Guericke University of Magdeburg, 39106 Magdeburg, Germany
[3] Department of Sport Science, Ruhr University Bochum, 44801 Bochum, Germany
* Correspondence: herbert.wagner@plus.ac.at

**Featured Application: The presented team handball PlayerScore can be applied to determine the individual player performance in team handball competition.**

**Abstract:** In team handball, the individual match performance of each player is essential for winning games; however, a validated match analysis system is still lacking. Consequently, the aim of the study was to justify (1) the different relevant variables and their scoring within the individual match analysis (PlayerScore), (2) to determine the intra-rater reliability and validity of the PlayerScore, and (3) to determine the influence of the rater in relation to their degree of expertise level. Six games (three games each of Spain and Brazil, one game twice) of the 2021 World Championship were analyzed by six different raters. The PlayerScore was calculated for each field player of Spain and Brazil in all seven analyzed games. We found a high intra-rater reliability (ICC = 0.97) for the two rated games (Spain against Germany), a highly significant difference ($p < 0.001$) between the summarized team PlayerScore of Spain and Brazil, as well as significant differences ($p < 0.001$) for the factor "game" and "rater" ($p < 0.05$), but no significant interaction for "game × rater" ($p = 0.90$) in the two-way repeated measures ANOVA. We conclude that the PlayerScore is a reliable and valid rating tool to determine the individual players' performance in team handball; however, the raters should have sufficient experience in the different techniques and tactics in team handball.

**Keywords:** performance; match statistics; intra-rater reliability; validity; rating

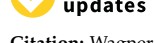



## 1. Introduction

In team handball, the individual match performance of each player is essential for winning games; however, measuring this individual match performance is a complex problem in sport science. In team handball training and match practice, the subjective coach's judgment is often used to analyze individual performance. However, the method of subjective rating of performance outcomes could contain a certain outcome bias that can negatively influence differing players' specific roles, varying players' involvements and the effects of their actions on the game, thus falsifying the data obtained and having negative effects on the scientific quality of the data [1,2]. It is known that coaches typically focus their attention on the critical aspects of the game, so most peripheral game actions are either ignored or remembered poorly [3,4]. There is always a danger of concentrating on useless or irrelevant performance indicators, which can lead to the formulation of trivial training recommendations or the derivation of insane training goals. The feedback derived from this is often inadequate for players and therefore not effective in terms of optimal training and performance control. For this reason, providing non-biased and objective feedback to players and coaches seems to be more efficient in order to be successful [5].

Depending on the major purposes of the match analysis (e.g., time and movement analysis, tactical analysis, technical analysis, collection of statistical data), the concept of performance indicators in the context of game analysis has proven its worth so far [6–8], whereas the integration of all dimensions of performance is important [9]. Relevant performance indicators are typically based on valid measurement methods. There have been numerous studies in team ball sports using different approaches to match analysis, and a great variety of performance indicators and calculation methods [10]. In team handball, match analysis can be classified into four different scopes: (i) descriptive analysis [11–13], (ii) comparative analysis, (defensive analysis, playing position, game results) [14–16], (iii) contextual variables (game location, time outs) [17–20], and (iv) predictive analysis (goal scoring, game results, goal keeper substitution, players' exclusion or additional field player) [21–27]. Most of these studies used game-related tournament statistics provided by the European Handball Federation (EHF) and International Handball Federation (IHF), or their own observational approaches, to measure performance indicators. Those studies using the match statistics provided by the EHF and IHF lose control of their own data, as the EHF and IHF do not provide detailed information on how they determine these statistics. Consequently, in scientific studies, self-performed match analysis should be preferred instead of match statistics provided by the EHF and IHF [28]. However, to the best of our knowledge, there are no scientific studies (i) which analyze individual player performances, exploiting the full range of situational technical actions in offensive and defensive sequences during play, (ii) which create an individual score system for an individual player based on the positive and negative effects of the single game actions made, and (iii) which mathematically take the factors of time and goal difference of a single game action into account, based on the current scientific state of research.

To counteract this lack of knowledge, the PlayerScore was developed to determine the individual player performance of each field player during the 26th Men's Handball World Championship in Denmark and Germany in 2019 [29]. The aim of the PlayerScore was to calculate only one significant variable regarding player performance, instead of numerous variables, as is common in a match analysis. The authors developed a ranking procedure based on the recorded actions per player in the game. The total score was calculated by the weighted sums of five positive actions (goal, assist, steal, block, and penalty received) and six negative actions (missed shot, technical fault, turnover, 2 min suspension, penalty fetched, and red card). It was found that there was a high correlation between the PlayerScore best-ranked players and the all-star-team, nominated by the IHF, as all of the PlayerScore first or second-ranked players in each playing position (left and right backcourt, center, left and right wing, pivot) were nominated in the all-star-team by the IHF. The player with the highest PlayerScore (144.53), Mikkel Hansen from Denmark, was also elected as the most valuable player of the tournament. The PlayerScore was also used in the 14th Men's Handball European Championship in Austria, Norway, and Sweden in 2020, but the results were not published in a scientific study. The positive and negative actions used for the PlayerScore are not a complete listing of all important actions during a match, but depend on the data provided by the IHF and EHF. Generally, the PlayerScore, as an objective feedback tool, was developed for coaches and players in elite team handball, not for scientific studies. Therefore, there is a lack of scientific validation of the PlayerScore. Consequently, the research objective of the present study is to optimize the PlayerScore by considering all important actions during a match, providing a clear justification of which actions were used and a science-based calculation of the score for these actions, applying the PlayerScore in an international elite team handball event by experienced raters, determining the reliability and validity of the player score, and providing perspectives on how the PlayerScore should be used as an objective match analysis tool in elite team handball.

Summarized, the aim of the study was (1) to justify the various PlayerScore relevant variables and their scoring within the PlayerScore calculation, (2) to determine the intra-rater reliability and validity of the PlayerScore, and (3) to determine the influence of the

rater in relation to their degree of expertise. We hypothesized a high intra-rater reliability, a significantly higher PlayerScore in the better-placed team, and differences in the PlayerScore due to the degree of expertise level of the raters.

## 2. Materials and Methods

### 2.1. Participants

For the validation of the PlayerScore, we selected two elite teams from the 27th Men's Handball World Championship in Egypt 2021. Only the best teams from each continent are qualified for this tournament, making them some of the best teams in the world. The selection of players for each national team was carried out by the coaching team before the tournament started, as determined by the IHF rules. Consequently, only those players nominated by each coaching team could be included in the study. Another inclusion/exclusion criteria of the study was to find two teams who performed similarly in face-to-face matches, who played in the same groups (preliminary and main round) but were placed far apart from each other in the final ranking of the tournament. The perfect match for these two teams was Brazil and Spain. Brazil and Spain played in the same preliminary (Group B) and main group (1st main group), drew (29:29) in the face-to-face match, while Spain won the bronze medal (3rd place) and Brazil was ranked 18th in the final standing of the tournament. To guarantee a high playing level, we selected only those matches of Spain and Brazil in which the goal difference at the end of the game was not higher than ten goals; consequently, the two matches against Uruguay (Spain-Uruguay 38:23 and Brazil-Uruguay 37:17) were not analyzed. In summary, six games (Brazil against Germany, Hungary, and Poland, as well as Spain against Germany, Hungary, and Poland) were selected for the measuring procedure.

All field players who played in these six matches (inclusion/exclusion criteria) were observed; consequently, 17 Spanish (five wings, eight backcourt players, and four pivots) and 15 Brazilian (four wings, eight backcourt players, and three pivots) team handball field players were analyzed in six matches. All matches were analyzed using the saved videos produced by the official TV broadcaster during the 27th Men's Handball World Championship in Egypt 2021.

In team handball, every professional player must sign an informed consent allowing their games to be broadcast live on television and for the videos to be used for statistical analysis. This informed consent is necessary because the European Handball Federation (EHF) and International Handball Federation (IHF) make the individual players' statistics of all teams and all games freely available online during international team handball Championships (European and World Championship). Consequently, an additional informed consent signed by the participants and the approval of an ethics committee was not necessary. However, the study complied with the requirements of the local Committee for Human Research Ethics, as well as current laws and regulations.

### 2.2. Study Design

The study was divided into preparation, rating, and calculation phases. In the preparation phase, we defined all relevant variables based on previous studies [12,13,16,21,22,25,26,28–30] and determined the different score of each variable based on the overall statistics of the 26th Men's Handball Word Championship in Denmark and Germany 2019, the 14th Men's Handball European Championship in Austria, Norway, and Sweden 2020, as well as the 27th Men's Handball Word Championship in Egypt 2021.

Before we began the rating process, inclusion criteria were defined for all raters. The minimum criteria for the raters was a coaching license in team handball and more than one year of experience as a team handball coach in an official competition. To determine the influence of the raters in relation to their degree of expertise, we defined three groups of two raters each, with varying levels of expertise:

Highest expertise level: Two EHF-Master Coaches (the highest coaching license in team handball), more than ten years of experience as a coach in an elite handball team

(national or international top level), a graduate degree in physical activity or sport science, and experience in observing and rating team handball matches.

Medium expertise level: Two A-License Coaches (the highest national coaching license), more than five years of experience as a coach in an experienced handball team, a graduate degree in physical activity or sport science, and experience in observing and rating team handball matches.

Low expertise level: Two coaches with a lower coaching license (B- or C-License Coach), more than one year of experience as a team handball coach in an official competition.

One EHF-Master Coach and two A-License Coaches had postgraduate master's degrees in sport science and experience in scientific studies.

In the rating phase, the six selected raters rated six matches (Brazil against Germany, Hungary, and Poland, as well as Spain against Germany, Hungary, and Poland) in a randomized order, except for the match Spain against Germany. To determine the intra-rater reliability of the PlayerScore, the match Spain against Germany was analyzed twice, always at the beginning (first game) and at the end (last game) of the rating process (to prevent remembering of the first and second rating). Consequently, all raters analyzed seven matches (six different matches and Spain against Germany a second time). An instructional video (typical playing sequences in the game Spain against Brazil) was produced, and the raters were taught how to rate the match, as well as all raters reading the PlayerScore Manual. The first author of the study was the contact person for all questions of the raters during the rating process. After familiarizing themselves with the rating procedures, the six raters analyzed all seven games using their own video analysis software (team handball coaches are familiarized in analyzing game videos utilizing their own software). All video analysis software was able to fast forward, rewind, stop, and play the video in slow motion.

In the calculation phase, the PlayerScores of all players were calculated, summarized, and then used for statistical analysis.

### 2.3. The PlayerScore

The basic consideration of the PlayerScore is that a goal increases the match score by one, so the score for a goal in PlayerScore is 1.0. The percentage for scoring a goal varies depending on the throwing position (Table 1); however, for the match score, it does not matter how the goal was scored (whether from 6 m or under hard contact from 10 m). The final action in team handball for scoring goals is a shot, and the last pass is defined as an assist [14,16,21]. To define the score for an assist, we used the overall statistics (available freely on the IHF and EHF homepage) from the 26th Men's Handball World Championship 2019 (IHF 2019), the 14th Men's Handball European Championship 2020 (EHF 2020), and the 27th Men's Handball World Championship (IHF 2021). A total of 23934 shots during these three tournaments resulted in 14634 goals (Table 2). The calculated scoring percentage was 61.1% (~60%); thus, the score for an assist, a pass from one offensive player to his teammate leading to a goal, is 0.60. If the assist leads to a penalty, it is rated as if it led to a goal.

$$\text{Assist} - \text{Goal} = \left( \frac{23,934}{14,634} \right) = 0.611 \approx 0.60$$

By definition, an assist is the last pass leading to a goal, but in team handball, the decisive action for a goal is sometimes not the last pass. If the center back player (playmaker) binds two defenders, creating a numerical superiority for the right attacking side, and his pass to the right backcourt player is directly forwarded as the last to the right wing player who scores a goal, then the assist was given by the right backcourt player, but the decisive action was by the playmaker. Consequently, in such cases, both backcourt players, the decisive player with the preceding action and the player with the last pass, get the score for an assist ("Assist—Goal"). The percentage for a shot without scoring a goal (missed, post, blocked or goalkeeper save) was ~40% (Table 1), so we defined the score for an assist without scoring a goal ("Assist—No goal") as 0.40. The positive score is due to the positive action of the assist, as it enables the teammate to score a goal, even if they do not.

**Table 1.** Calculation of the different scores based on the percentage (%) of the overall statistics of the 26th Men's Handball Word Championship 2019 (IHF 2019), the 14th Men's Handball European Championship 2020 (EHF 2020), as well as the 27th Men's Handball Word Championship (IHF 2021).

| | All Shots | | Fast Break | | Shot 9 m | | Breakthrough | | Penalty | | Shot 6/7 m | | Block | |
| | Goal | Shot | Goal | Shot | Goal | Shot | Goal | Shot | Goal | Shot | Goal | Shot | Block | No Goal |
|---|---|---|---|---|---|---|---|---|---|---|---|---|---|---|
| IHF 2019 | 5239 | 8694 | 869 | 1085 | 1231 | 2947 | 605 | 812 | 519 | 686 | 4008 | 5747 | 404 | 1716 |
| EHF 2020 | 3532 | 5779 | 420 | 518 | 934 | 2009 | 402 | 547 | 316 | 434 | 2557 | 3722 | 273 | 1075 |
| IHF 2021 | 5863 | 9461 | 791 | 1031 | 1231 | 2947 | 865 | 1146 | 568 | 744 | 4409 | 6181 | 398 | 1716 |
| | 14,634 | 23,934 | 2080 | 2634 | 3396 | 7903 | 1872 | 2505 | 1403 | 1864 | 10,974 | 15,650 | 1075 | 4507 |
| Percentage | 61.1 | | 79.0 | | 43.0 | | 74.7 | | 75.3 | | 70.1 | | 23.9 | |
| **Score** | **0.60** | | **0.80** | | **0.45** | | **0.75** | | **0.75** | | **0.70** | | **0.25** | |

**Table 2.** Definition of the PlayerScore playing actions.

| Variable (Number) | Score | Definition |
|---|---|---|
| Goal (1) | 1.00 | Scoring a goal. The score increases by one > Score 1.0 |
| Suspension fetched (2) | 0.80 | 2 min suspension or red card fetched by an offensive player. In the EHF Champions League 2020 the difference in goals between the teams in numerical inferiority or superiority was ~0.80 > Score 0.80 |
| Pivot play (3) | 0.75 | Successful back running (behind the defensive player), a direct or indirect block leading to a direct shot, assist, penalty or breakthrough of the wing or backcourt player. The breakthrough percentage in the international tournaments IHF 2019/2021 and EHF 2020 was 75% > Score 0.75 |
| Winning one-on-one (4) | 0.75 | Winning one-on-one of the wing, backcourt player or pivot (direct or indirect block of the pivot) leading to a direct shot or assist (only the last action counts). The breakthrough percentage in the international tournaments IHF 2019/2021 and EHF 2020 was 75% > Score 0.75 |
| Screen (5) | 0.45 | Screen of an offensive player to enable a free 9 m shot. The 9 m shot scoring percentage in the international tournaments IHF 2019/2021 and EHF 2020 was 45% > Score 0.45 |
| Penalty fetched (7) | 0.75 | Penalty fetched by an offensive player. The penalty percentage in the international tournaments IHF 2019/2021 and EHF 2020 was 75% > Score 0.75 |
| Assist—Goal (8) | 0.60 | Pass from one offensive player to another leading to direct goal. The scoring percentage in the international tournaments IHF 2019/2021 and EHF 2020 was ~60% > Score 0.60 |
| Assist—No goal (9) | 0.40 | Pass from one offensive player to another without scoring a goal (missed, post, block or goalkeeper save). The missing percentage in the international tournaments IHF 2019/2021 and EHF 2020 was ~40% > Score 0.40 |
| Steal (11) | 0.80 | Turnovers by snatching the ball due to optimal anticipation of the defense player. A steal is usually leading to a fast break. The fast break percentage in the international tournaments IHF 2019/2021 and EHF 2020 was 80% > Score 0.80 |
| Block (12) | 0.25 | Defensive block by the defense player after 9 m shot of the rival offensive player. Percentage of blocks relative to all missing shots in the international tournaments IHF 2019/2021 and EHF 2020 was 25% > Score 0.25 |
| Offensive foul fetched (13) | 0.60 | Provoked offensive foul by the defensive player. The turnover enables a following offensive action, whereas the scoring percentage in the international tournaments IHF 2019/2021 and EHF 2020 was 60% > Score 0.60 |
| Suspension received (22) | −0.80 | 2 min suspension or red card received by a defensive player (no coaching staff). In the EHF Champions League 2020 the difference in goals between the teams in numerical inferiority or superiority was ~0.80 > Score 0.80 |
| Technical error (23) | −0.60 | Turnovers made by an offensive player due to an offensive foul, intercepted pass, step error or time play. The turnover enables a following offensive action, whereas the scoring percentage in the international tournaments IHF 2019/2021 and EHF 2020 was 60% > Score 0.60 |

**Table 2.** *Cont.*

| Variable (Number) | Score | Definition |
|---|---|---|
| Losing one-on-one (24) | −0.75 | Losing a one-on-one against the rival wing, backcourt player or pivot (direct or indirect block of the pivot) leading to a throw or assist (only the last action counts). Losing a one-on-one normally leading to a breakthrough. The breakthrough percentage in the international tournaments IHF 2019/2021 and EHF 2020 was 75% > Score −0.75 |
| Shot 6/7 m—No goal (26) | −0.70 | 6 m shots, wing shots, breakthrough, fast break or penalty (7 m) without scoring a goal (missed, post or goalkeeper save). Even if the ball remains in possession of the attack. The scoring percentage in the international tournaments IHF 2019/2021 and EHF 2020 was ~70% > Score −0.70 |
| Penalty received (27) | −0.75 | Penalty received by a defensive player. The penalty percentage in the international tournaments IHF 2019/2021 and EHF 2020 was 75% > Score −0.75 |
| Shot 9 m—No goal (29) | −0.45 | 9 m shots (the area from a backcourt player either over or through the defense, and after breakthrough but with another defense player in front) without scoring a goal (missed, post, block or goalkeeper save). Even if the ball remains in possession of the attack. The 9 m shot scoring percentage in the international tournaments IHF 2019/2020 and EHF 2020 was 45% > Score −0.45 |

As shown in Table 1, the scoring percentage is different due to the throwing position. The highest percentage was found in fast breaks (79.0%), followed by penalties (75.3%), breakthroughs (74.5%), and shots from 9 m (43.0%). The percentage for 6/7 m shots (fast break, breakthrough, penalty, close distance, and wing shots) was 70.1%. For the Player-Score, we separated 6/7 m and 9 m shots. [13,14,16,21,22]. Depending on the calculated scoring percentage we defined the score for a fast break, breakthrough, penalty (7 m), close distance (6 m) and wing shot without scoring a goal with −0.70 ("Shot 6/7 m—No goal") and the score for a 9 m shot (the area from a backcourt player either over or through the defense, and after breakthrough but with another defense player in front) with −0.45 ("Shot 9 m—No goal"). The player will get a score for a "Shot 6/7 m—No goal" or "Shot 9 m—No goal" even if the ball remains in possession of the attack (rebound).

A penalty (7 m throw) in team handball is awarded when a clear chance of scoring is illegally destroyed on the court by a player of the opposing team [31]. The scoring percentage of a penalty was 75.3% (Table 1); consequently, we defined the score for a received penalty by a defensive player with −0.75 ("Penalty received") and for a penalty fetched by an offensive player with 0.75 ("Penalty fetched"). A 2 min suspension in team handball is awarded for fouls that are committed with high intensity or against an opponent who is running fast, clasping on to the opponent for a long time or pulling him down, fouls against the head, throat or neck, hard hitting against the torso or throwing arm, attempting to make the opponent lose body control, and running or jumping with great speed into an opponent [31]. In the EHF Champions League 2020 the difference in goals between the teams in numerical inferiority and superiority was ~0.80; consequently, we defined the score for a received suspension by a defensive player with −0.80 ("Suspension received") and for a suspension fetched by an offensive player with 0.80 ("Suspension fetched"). A direct red card was rated like a 2 min suspension because the consequence is the same (2 min inferiority or superiority for the team) [26,27]. A 2 min suspension or a red card for a team official was not rated, because the subsequent numerical minority was not caused by the action of a player.

To consider all important actions in defense (stealing, blocking, receiving an offensive foul, winning or losing a one-on-one), we defined the following variables for the Player-Score [12,14–16,21,22,26–29]. A steal is defined as a turnover by snatching the ball due to optimal anticipation of the defensive player. A steal usually leads to a direct fastbreak, and the throwing percentage in the fast breaks was ~80% (Table 1); consequently, we defined the score for the steal by a defensive player as 0.80 ("Steal"). A block is defined as a defensive block by the defensive player during a 9 m shot of the rival offensive player. The percentage of all blocks, relative to all shots without scoring a goal, was ~25% (Table 1); consequently,

we defined the score for the block of a defensive player as 0.25 ("Block"). A provoked offensive foul by the defensive player leads to a turnover. The turnover enables a following offensive action, and the overall throwing percentage in offence was ~60% (Table 1); consequently, we defined the score for received offensive foul as 0.60 ("Offensive foul received"). An essential tactical component in team handball is a one-on-one action. In defense, the defensive player tackles the offensive player by blocking the opponent with arms, hands, legs, or uses any part of the body to displace him or push him away as well as hold him. Winning this one-on-one leads to a foul without personal punishment [31], and interrupts the offensive action. If a defensive player wins the one-on-one or one-on-two in an isolated situation (no support from other defensive players), the player will normally prevent a breakthrough of an offensive player, and the throwing percentage after a breakthrough was ~75% (Table 1); consequently, we defined the score for this action of a defensive player as 0.25 ("Winning one-on-one (defense)"). Losing a one-on-one against the rival wing, backcourt player or pivot (direct or indirect block of the pivot) leading to a shot, assist (only the last action counts) or penalty was scored with −0.75 ("Losing one-on-one (defense)") due to the throwing percentage of ~75% for a breakthrough (Table 1).

To consider all important actions in offense (winning one-on-one, screening, blocking, or running behind, as well as technical errors), we defined the following variables for the PlayerScore [12,14–16,21,22,26–29]. Winning a one-on-one of the wing or backcourt player leading to a shot, assist (only the last action counts), or penalty was scored with 0.75 ("Winning one-on-one (offense)") due to the throwing percentage of ~75% for a breakthrough (Table 1). A screen that enables a free 9 m shot of a backcourt player was scored with 0.45 ("Screen") due to the throwing percentage of ~45% for a 9 m shot (Table 1). For the pivot, we defined separate variables because a pivot has a different tactical profile in the offense in comparison to wings or backcourt players. Successful back running (behind the defensive player), a direct or indirect block leading to a direct shot, assist, penalty, or breakthrough of the wing or backcourt player was scored with 0.75 ("Pivot action") due to the throwing percentage of ~75% for a breakthrough (Table 1). Finally, we rated turnovers made by an offensive player due to an offensive foul, intercepted pass, step error, or time play. The turnovers enable a following offensive action, so we defined the score for this action of an offensive player as −0.60 ("Technical error").

To avoid overrating the different actions, we counted only the last action [29]. For example, if a defensive player lost a one-on-one action against the rival backcourt player leading to a penalty, we counted only the last action ("Penalty fetched"). However, if the defensive player also got a 2 min suspension, both actions ("Penalty fetched" and "Suspension fetched") were counted, because both actions decrease the match performance (penalty for the rival team and 2 min minority for the teammates). In Table 2, we depicted all playing actions determining the PlayerScore.

In team handball competition, the physical load increases with playing time due to an increase in blood lactate concentration [32–34], muscular fatigue (maximal voluntary contraction and rate of force development) [35], and mental fatigue [36]. Consequently, the technical and tactical requirements increase with playing time, which was taken into account by the time factor (Figure 1A). The time factor was defined as a linear function [29] and increased depending on the playing time (from 0.5 in the first to 1.5 in the last minute). In addition to the physical load, performance in team handball is influenced by mental loading pressure (e.g., choking under pressure) [37] and emotional stability in critical situations [36], such as a tight score. Consequently, the match score (goal difference) was taken into account in the PlayerScore by a bell function (2.0 for a goal difference of +1, 0 and −1; 0.1 for a goal difference of +16 and −16, etc.) as shown in Figure 1B.

*2.4. Procedures*

In the rating process, the raters registered the playing score, playing time, number of players, and the number of every playing action in an Excel data sheet (Microsoft Excel 365, Microsoft Corporation, Redmond, Washington, DC, USA) (Figure 1, rating data sheet).

After finishing the rating process, the PlayerScores of all players were calculated using Microsoft Excel and stored on a separate Excel data sheet. In this calculation process, the score (negative or positive) of every registered playing action was multiplied with the time and goal difference factor, accumulated, and stored separately per player in the rows and per player action in the columns (Figure 2, PlayerScore data sheet). The final PlayerScore per player was the sum of all scores and was calculated in a separate column. This calculation process was repeated for all matches and all raters and stored on different data sheets. Finally, all PlayerScores of all players and matches, separated by the six raters, were summarized in an Excel sheet (Figure 2, PlayerScore summarized data sheet) and used for statistical analysis.

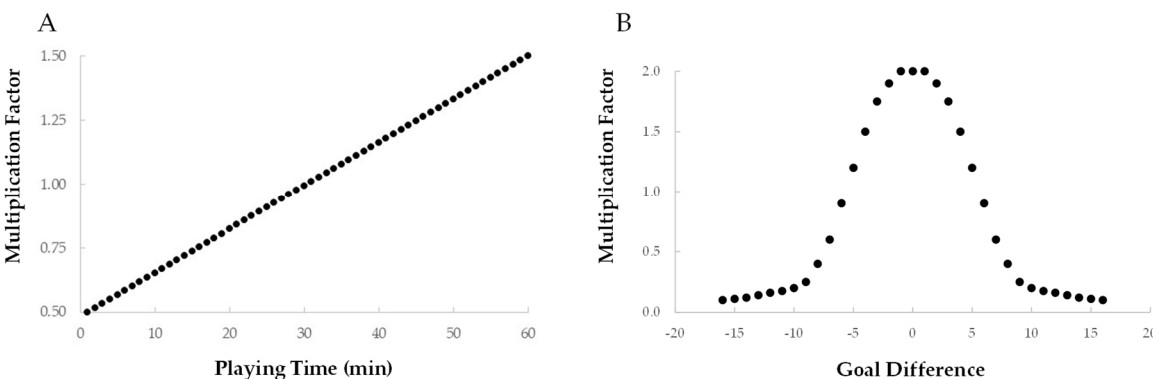

**Figure 1.** Linear function (**A**) for the playing time factor and Gaussian bell function (**B**) for the goal difference factor in the PlayerScore.

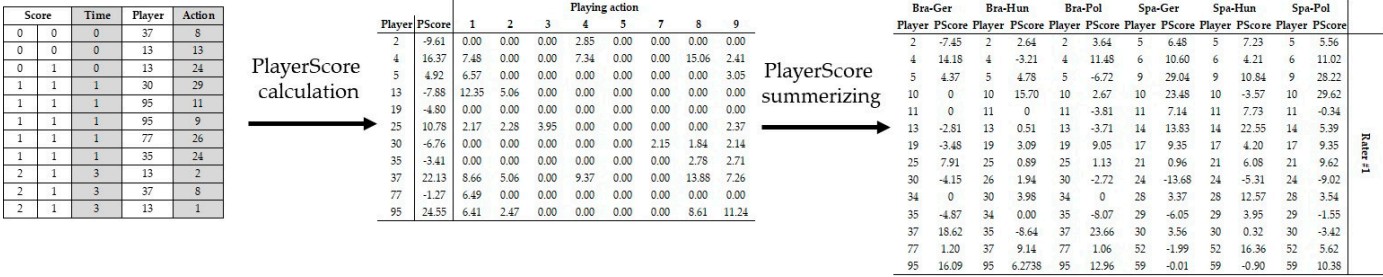

**Figure 2.** Outline of the methodology for the PlayerScore rating, calculation and summarizing process (PScore: PlayerScore; Bra-Ger: Brazil against Germany; Bra-Hun: Brazil against Hungary; Bra-Pol: Brazil against Poland; Spa-Ger: Spain against Germany; Spa-Hun: Spain against Hungary; Spa-Pol: Spain against Poland).

*2.5. Statistical Analyses*

All statistical analyses were conducted using SPSS ver. 27 (IBM Corp., Armonk, NY, USA) with a *priori* significance of $p < 0.05$ for all tests [38,39]. Mean values ± standard deviations and 95% confidence intervals of the variables were calculated for descriptive statistics. Normality of the data was verified by the Shapiro-Wilk test, and normality was found for all used variables. Intraclass correlation coefficient (ICC) (1,2), 2-way random effects model with single measure was calculated to determine the intra-rater (test-retest) reliability. A cut-off value of 0.70 was defined for the ICC. Linear regression was additionally calculated and diagrammed as linear regression analysis plots between the first and second rated games Spain against Germany. A paired sample T-test was calculated to determine the differences in the PlayerScore team performances between Brazil and Spain. Cohen's $d_z$ effect size was additionally calculated for the T-test. Finally, a two-way repeated-measures ANOVA with "rater" and "game" as the main factors was calculated. A Bonferroni post

hoc test was utilized to determine significant differences between the raters, and effect size ($\eta^2$) and power (1-ß) were calculated. For all statistical analyses, significance was set at $p < 0.05$ and the cut-off value for the effect size (large effect size) was defined as 0.14 for $\eta^2$ and 1.20 for Cohen's d [38,39].

## 3. Results

A linear regression plot of the PlayerScore team statistics between the first and second rating of Spain against Germany, as well as a bar graph of the PlayerScore team performance of Brazil and Spain, is shown in Figure 3. We found a high test-retest reliability (ICC = 0.97) between the twice-rated matches of Spain against Germany. The summarized team PlayerScore was 39.8 ± 18.1 for Brazil and 87.5 ± 20.2 for Spain, and the paired sample T-test revealed a highly significant difference ($p < 0.001$, Cohen's d = 19.2) between these two teams for their respective matches against Germany, Poland, and Hungary.

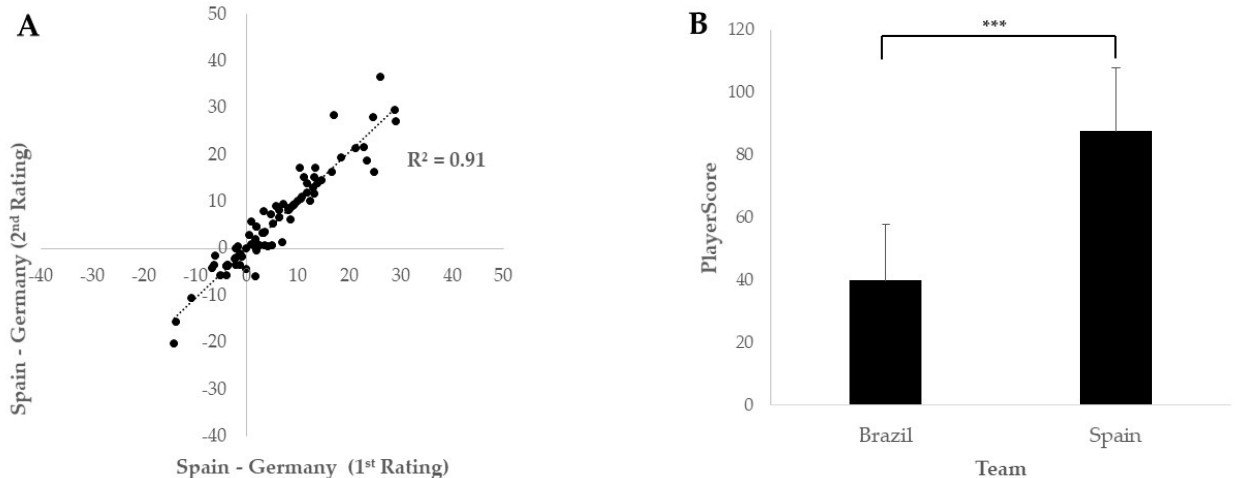

**Figure 3.** Linear regression plot of the PlayerScore team statistics between the first (1st) and second (2nd) rating of Spain against Germany (**A**), and bar graph of the PlayerScore team performance of Brazil and Spain (**B**). ($p < 0.001$ ***).

Descriptive data including means, standard deviations (±SD) and 95% confidence intervals for the PlayerScores, separated by the raters and games, are depicted in Table 3. In the repeated measures two-way ANOVA, we found significant differences for the factor "game" ($p < 0.001$, F = 8.55, $\eta^2 = 0.37$, 1-ß = 1.00) and "rater" ($p = 0.03$, F = 2.66, $\eta^2 = 0.15$, 1-ß = 0.78), but no significant interaction for "game × rater" ($p = 0.90$, F = 0.67, $\eta^2 = 0.04$, 1-ß = 0.41). The Bonferroni post hoc test revealed a significant difference ($p = 0.046$) only between Rater #5 and #6. For a detailed discussion of the results, the PlayerScore (mean value) for each player of both teams was additionally calculated. The players with the highest PlayerScore of both teams were from Spain the playmaker (55.7), the right backcourt player (42.9), the right wing (39.4) and from Brazil the left (35.9) and the right backcourt player (26.5).

**Table 3.** Mean value, standard deviation (±SD) and 95% confidence intervals (95% CI) for the PlayerScores, separate by the six raters and six games.

| Game | Rater #1 Mean ± SD (95% CI) | Rater #2 Mean ± SD (95% CI) | Rater #3 Mean ± SD (95% CI) | Rater #4 Mean ± SD (95% CI) | Rater #5 Mean ± SD (95% CI) | Rater #6 Mean ± SD (95% CI) |
|---|---|---|---|---|---|---|
| Brazil–Germany | 2.83 ± 8.29 (−1.96–7.62) | 2.04 ± 6.23 (−1.56–5.64) | 3.77 ± 11.98 (−3.15–10.68) | 2.37 ± 7.67 (−2.06–6.79) | 4.55 ± 6.40 (0.85–8.25) | 4.39 ± 6.64 (0.55–8.22) |

**Table 3.** *Cont.*

| Game | Rater #1 Mean ± SD (95% CI) | Rater #2 Mean ± SD (95% CI) | Rater #3 Mean ± SD (95% CI) | Rater #4 Mean ± SD (95% CI) | Rater #5 Mean ± SD (95% CI) | Rater #6 Mean ± SD (95% CI) |
|---|---|---|---|---|---|---|
| Brazil–Poland | 2.65 ± 5.66 (−0.62–5.92) | 1.26 ± 4.24 (−1.19–3.70) | 3.23 ± 9.64 (−2.33–8.80) | 1.39 ± 4.69 (−1.32–4.10) | 4.10 ± 4.56 (−1.46–6.73) | 2.77 ± 3.48 (0.76–4.77) |
| Brazil–Hungary | 2.90 ± 8.74 (−2.14–7.95) | 1.22 ± 5.77 (−2.11–4.55) | 2.40 ± 8.04 (−2.24–7.04) | 1.99 ± 6.56 (−1.80–5.78) | 5.09 ± 6.74 (1.20–8.98) | 5.72 ± 10.28 (−0.21–11.66) |
| Spain–Germany | 6.15 ± 11.10 (−0.26–12.56) | 4.64 ± 8.53 (−0.28–9.56) | 8.96 ± 13.81 (0.99–16.93) | 6.87 ± 11.89 (0.01–13.74) | 12.48 ± 9.73 (6.86–18.10) | −0.87 ± 5.30 (−3.93–2.19) |
| Spain–Poland | 7.43 ± 10.78 (1.20–13.65) | 6.69 ± 8.78 (1.62–11.76) | 11.27 ± 16.01 (2.03–20.51) | 7.83 ± 12.85 (0.41–15.25) | 15.68 ± 10.82 (9.43–21.93) | 2.66 ± 6.32 (0.99–6.31) |
| Spain–Hungary | 6.16 ± 7.66 (1.74–10.58) | 5.28 ± 7.07 (1.74–10.58) | 8.38 ± 8.82 (3.29–13.48) | 5.63 ± 7.12 (1.51–9.74) | 8.73 ± 8.46 (3.85–13.62) | 4.59 ± 8.04 (−0.05–9.23) |

The mean values of the PlayerScore team performance, separated by the raters and games, are shown in Figure 4. It is obvious that the PlayerScores differentiate significantly between all three matches of Brazil and Spain. However, it is also shown that Raters #1, #2, #3, #4, and #6 rated similarly in all six games, whereas the PlayerScore of Rater #5 was significantly different from all other raters.

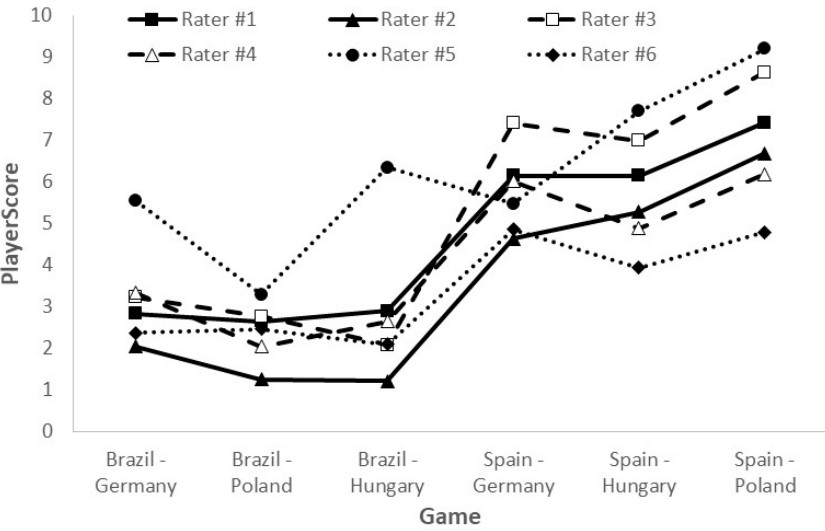

**Figure 4.** PlayerScore team performance (mean values) separated by raters and games.

## 4. Discussion

The aim of the study was to determine the intra-rater reliability and validity of the PlayerScore, as well as the influence of the rater's degree of expertise level. As expected, we found a high intra-rater reliability, determined by the test-retest reliability (ICC = 0.97) in the twice-rated matches of Spain against Germany. This high intra-rater reliability was also found in a similar study determining the validity and reliability in a live match analysis in soccer [40]. As shown by the linear regression plot of the PlayerScore team statistics between the first and second rating of Spain against Germany (Figure 3A), most players were rated very similarly, if not identically. We suggest that the raters were able to replicate their ratings very well after familiarizing themselves with the PlayerScore rating procedure. Consequently, the PlayerScore is a reliable rating tool for determining the individual players' performance in team handball.

To determine the validity, whether the better-placed team in the final ranking of the tournament had a higher PlayerScore compared to the worst-placed team was analyzed.

The paired sample T-test revealed a highly significant difference ($p < 0.001$) between the summarized team PlayerScore of Brazil and Spain in their matches against Germany, Poland, and Hungary. The calculated very large effect size (Cohen's d = 19.2) supported the conclusion that the PlayerScore is well-suited to determine the match performance in team handball and to discriminate between teams and players due to their performance in the team handball match. These results agree with the results of a previous study utilizing a simplified version of the PlayerScore in the IHF 2019 [29], because they also found a high difference in the PlayerScore between Denmark (PlayerScore: 51.1), who won the final 31:22 against Norway (PlayerScore: 16.7). Additionally, the authors also found a high relationship between the PlayerScore best-ranked players and the all-star team, nominated by the IHF. The player with the highest PlayerScore (144.5) in the IHF 2019, Mikkel Hansen from Denmark, was also elected for the MVP of the tournament [29]. In the present study, we have not analyzed all matches, but in the three analyzed matches of Spain and Brazil, the players with the highest PlayerScores in both teams were the dominant players of their teams in the whole tournament. The right wing of Spain, with a PlayerScore of 39.4 in the present study, was also nominated for the all-star team by the IHF. We suggest that the PlayerScore is a valid rating tool to determine the individual players' performance in team handball.

However, the PlayerScore used for the IHF 2019 and 2021, as well as the EHF 2020, was a cooperation project between one author of the present study and the IHF and EHF. Consequently, only those parameters (positive actions: goal, assist, steal, block, and penalty fetched; negative actions: missed shot, technical fault, turnover, 2 min suspension, penalty received, and red card) made available by the IHF and EHF were used to calculate the PlayerScore. For an extensive determination of all relevant actions in team handball, we added further positive (pivot play, winning one-on-one, screen, offensive foul, and suspension fetched) and negative (technical error and losing one-on-one) actions, as well as a more extensive differentiation (assist leading to a goal or without scoring, and missed shots from 6/7 m or 9 m) in our study. Consequently, the rating process was more difficult, and we had more actions (eleven positive and six negative) for calculating the PlayerScore. Additionally, the score was different from the previous PlayerScore determination [29,30]. However, comparing the summarized team PlayerScores in the present study (39.8 ± 18.1 for Brazil and 87.5 ± 20.2 for Spain) with the previous PlayerScore determination (1.1 ± 9.1 for Brazil and 49.8 ± 11.1 for Spain), we found a similar difference (47.7 versus 48.7) between Brazil and Spain due to the additional actions and the different score. Players who played more in defense (winning one-on-one and offensive foul fetched), and pivot players (pivot play and suspension fetched) performed better, often utilizing combined positive and negative actions and a different scoring in the present study. We suggest that the PlayerScore determination utilized in the present study is more suitable for determining the individual performance of each player, independent of their playing position and whether the player plays more in offense or defense.

Comparing the PlayerScores determined by the different raters, we found a quite similar rating of Rater #1, #3 and #4, a lower rating of Rater #2 and #6, and a significantly different rating of Rater #5. In the repeated measures two-way ANOVA, we also found a significant difference for the factor "rater", not only for the factor "game". In this context, it is interesting that Rater #1 (EHF-Master Coach), Rater #3 and #4 (A-License Coaches) have postgraduate master's degrees in sport science and experience in scientific studies, whereas Rater #2 (EHF-Master Coach) has only a graduate degree in physical activity and experience in observing and rating team handball matches, and Rater #6 (B-License Coach) has no graduate degree and only experience in team handball coaching. Rater #5, who rated completely differently, has the lowest coaching license (C-License Coach) and little experience in game analyses. We suggest that raters should have sufficient experience in the different techniques and tactics in team handball (EHF-Master or A-License Coach) and sufficient experience in the rating process. Especially in match sequences where many players are involved, it is often not easy to determine who has won or lost a one-on-one,

who has fetched or received a 2 min suspension, who has fetched a penalty, or who has blocked a 9 m shot. During international tournaments, the game statistics are always determined by several people to avoid errors; consequently, we suggest that a match should always be rated by one rater, checked by a second rater, and unclear scenes should be solved and rated together to avoid errors in the determination of the PlayerScore. The rating procedure (by only one rater) might be a limitation of the study because a more accurate rating (as described before) might lead to fewer differences in the PlayerScores between the different raters. Another limitation of the study is that we have only analyzed the PlayerScore and no comparable variables; however, in this study, the aim was to optimize and validate the PlayerScore. To determine the influence of general (strength, endurance, agility, etc.) [32,34,36,41] or specific physical performance (team handball game-based performance) [32–34] on the match performance (PlayerScore), additional studies are warranted.

To compare the results of the present study with previous studies in team handball match analysis was not possible because the methods were quite different. Whereas in the previous studies [12–16,18,21,22,24–28] the players' match performance was determined by several variables such as the number or efficiency of attacks, shots (court position and/or goal area), assists, technical faults, blocks, fastbreaks, breakthroughs, steals, goalkeeper or field player defense, 2 min suspension, etc., the PlayerScore is only one overall performance factor including all of these different variables. Consequently, it is much easier to compare different players within one team or different teams (who performed best in competition), or to evaluate the development (e.g., an increase or decrease in the PlayerScore) of one or more players within a playing period. In this context, it might also be interesting to determine the relative PlayerScore (per minute played in the match) to measure the match performance independent of the total playing time.

## 5. Conclusions

The aim of the study was to determine the intra-rater reliability and validity of the PlayerScore, as well as the influence of the rater's degree of expertise. We concluded that the PlayerScore is a reliable and valid rating tool for determining individual players' performance in team handball. Due to the calculation of the PlayerScore (which uses only one overall performance factor instead of several variables), the PlayerScore is a much more practical tool for match analysis in team handball compared to previous methods. We recommend the use of the PlayerScore in international team handball tournaments, National Team Handball Leagues, or for receiving objective feedback about individual players' performances from team handball coaches in their respective teams.

**Author Contributions:** Conceptualization, H.W., M.H., K.M. and V.R.; methodology, H.W., M.H., V.R. and J.U.; validation, H.W., M.H. and V.R.; formal analysis, H.W. and J.U.; investigation, H.W., M.H. and J.U.; resources, M.H. and K.M.; writing—original draft preparation, H.W.; writing—review and editing, H.W., M.H., K.M. and J.U.; visualization, H.W., M.H. and J.U.; supervision, H.W. and K.M.; project administration, M.H. and K.M.; All authors have read and agreed to the published version of the manuscript.

**Funding:** This research received no external funding.

**Institutional Review Board Statement:** The study complied with the requirements of the local Committee for Human Research Ethics, as well as current laws and regulations.

**Informed Consent Statement:** Informed consent was obtained from all subjects involved in the study.

**Data Availability Statement:** The data presented in this study are available on request from the corresponding author. The data are not publicly available due to data protection rules of the University of Salzburg.

**Acknowledgments:** We want to thank all raters for the time and enthusiasm during the rating process.

**Conflicts of Interest:** The authors declare no conflict of interest.

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
