# Peer review of "The PlayerScore: A Systematic Game Observation Tool to Determine Individual Player Performance in Team Handball Competition"

_applsci, doi:10.3390/app13042327_

Round 1
Reviewer 1 Report
Dear Authors,
The the aim of the study was (1) to justify the different PlayerScore relevant variables and its scoring within the PlayerScore calculation, (2) to determine the intra-individual reliability and validity of the PlayerScore, and (3) to determine the influence of the rather in 150 relationship to their degree of expertise. It is an interesting manuscript with an aim with a research gap (instrument validation), however some sections need some major revisions.
Please, see the following point-by-point revisions in attachment.
Good work!

Author Response
Reviewer #1:
General Comments
The the aim of the study was (1) to justify the different PlayerScore relevant variables and its scoring within the PlayerScore calculation, (2) to determine the intra-individual reliability and validity of the PlayerScore, and (3) to determine the influence of the rather in 150 relationship to their degree of expertise. It is an interesting manuscript with a strong research gap and design (instrument validation), however some sections need some major revisions.
A: We would like to thank you for your helpful comments. We have attempted to follow all of your suggestions and have revised our manuscript to the best of our ability. Below, you will find detailed responses (point by point) to your review.
Specific comments
Please, consider the following point-by-point revisions:
Lines 13–15 (Featured Application): Please, place this section after the conclusions
of the study. It doesn't seem within the author instructions (MDPI).
A: We used the Word template provided by the journal homepage, and in this template, the featured application is at the beginning (before the abstract) and not after the conclusion of the manuscript.
· Lines 16–42 (Abstract): The summary is too large (i.e., reduce from 294 words to 200 words).
A: As per the reviewer's suggestion, we reduced the abstract (207 words, L16-36).
· Line 47–68 (Introduction): Some phrases are without literature support (lines 47 to 50, 50 to 53). I would suggest framing the research topic in the very first sentences of the introduction (i.e., individual player’s performance. The first two sentences are common sense. Actually, the study object is the individual performance, and research aim is to validate the PlayerScore system.
A: As per the reviewer's suggestion, we rephrased the first sentences focusing on the individual player's performance. Consequently, your comments regarding the literature support are no longer relevant. (L40-45).
· Lines 73 – 77 (Introduction: Please, add the importance of integrating all dimensions of performance (e.g., time and movement analysis, tactical analysis, technical analysis, collection of statistical data) (please, see: https://peerj.com/articles/14381/).
A: As per the reviewer's suggestion, we added this information and also the reference in the resubmission (L70-71).
· Lines 69–84 (Introduction): Also, there are key points that must be instead of feedback (learning factor). Match analysis using observational approaches to measure performance indicators (https://doi.org/10.3389/fpsyg.2019.02077).
A: Based on your comments and those of the other reviewers, we reduced this paragraph and focused more on observational approaches to measure performance indicators (L61-91).
· Line 144 (Introduction): Figure should be placed immediately after immediately after
its mention in the text.
A: Based on the suggestion of Reviewer #6, we placed the entire paragraph, including the figure, at the end of the Methods section (L369-384).
· Line 150 (Introduction): Some English corrections are required. (e.g., “Rather” instead “Rater”).
A: A native speaker corrected the entire manuscript before resubmitting it.
- Lines 157 –175 (Participants): What selection criteria were used to include Spanish
and Brazilian team handball? What is the context (i.e., elite, sub elite, amateur)?
A: We have explained the selection criteria in the study design; however, we agree with the reviewer's suggestion to include this paragraph at the beginning of the "Participants" section (L136-178).
- Lines 177 – 226 (Study design): What recommendations or guidelines did you base
your sample and procedure selection on? (please, see research line of Anguera and
collaborators (please: http://dx.doi.org/10.5672/apunts.2014-983.cat.(2018/4).134.08
or https://doi.org/10.3389/fpsyg.2019.01476)
A: In the resubmission, we have clarified the selection process. We have also read the papers you suggested, but the observation tool used in this study is quite different from the methods used in our study (L188-209).
Line 277 (The team handball PlayerScore): Please, add references about research
procedures (i.e., PlayScore, procedures, statistical analysis). The methodological
procedures should be replicable by other researchers.
A: As per your suggestion and the suggestion of Reviewer #4, we added references about the research procedures and an outline of the methodology for the PlayerScore rating, calculation, and summarizing process (L180-382).
· Lines 400 – 402 (Statistical analyses): Please, add references (e.g., Hopkins, Cohen, etc.; please, see: doi:10.1249/MSS.0b013e31818cb278). Also, add cut-offs values for effect size (η2) and Intraclass correlation coefficient (ICC). Cohen d should be considered for t-test.
A: As suggested by the reviewer, we added these references, information about the cut-off values to the methods (L408-426), and Cohen's d to the results (L435).
· Lines 445 (Discussion): The results lack a relationship to contextual data, something you report in the introduction. Does the validity of PlayerScore hold? If you are unable to report this relationship, you should report the future analysis perspective. Contextual variables highly affect the individual match running performance (https://www.mdpi.com/2075-4663/10/8/121).
A: We are certain that the validity of the PlayerScore holds; however, we agree with the reviewer and the other reviewers that we should add more results. As a result, we added the PlayerScore of the top five ranked players of both teams to the results (L450-455) and also to the discussion to support the validity of the PlayerScore (L501-506).
A poor scientific writing was found in discussion section. Please, make an extensive correction of English style and form (as well as the use of symbols such as #).
A: A native speaker corrected the entire manuscript before resubmitting it; however, using the hashtag symbol for numbering the raters is common in scientific studies.
· Lines 493 – 495 (Discussion): Is PlayScore better than other methods available in the literature? Why should a handball coach/analyst use it? Please, add frame the validated model with other methods present in the literature.
A: You, as well as reviewers #4 and #6, are absolutely right; the discussion needs more information about the advantages of the PlayerScore compared to other match analysis methods. Consequently, we have added an additional paragraph to the resubmitted manuscript (L572-586).
· Lines 518 – 529 (Conclusions): Conclusions should be short and concise. Likewise,
they should not report conclusions that they have not actually studied (i.e. strength,
endurance, and agility).
A: We have reworded the conclusion and removed the part about strength, endurance, and agility (L588-599).
· Lines 544 – 580 (References): The work is poor in bibliographical support (especially the introduction and methodology). Please, add references until more or less 30.
A: As suggested by the reviewer, as well as other reviewers of the manuscript, we added more references. In the resubmitted manuscript, we have 41 references (L613-703).

Reviewer 2 Report
Please improve the literature review. Method should be clearly discussed. Discussion and conclusion needs more to analysis.
Author Response
Reviewer #2:
Please improve the literature review.
A: Based on your suggestion and the suggestions of the other reviewers, we improved the introduction, including the literature review (L61-91).
Method should be clearly discussed.
A: Based on your suggestion and the suggestions of the other reviewers, we improved the methods (L137-426).
Discussion and conclusion needs more to analysis.
A: Based on your suggestion and the suggestions of the other reviewers, we improved the discussion (L486-586) and conclusion (L588-599), added some results, and included an additional paragraph (L572-585).

Reviewer 3 Report
interesting work is thought that this research can be a performance determination criterion in athletes. however, more research is needed. it is also necessary to integrate different tests that determine performance.
Author Response
Reviewer #3:
Interesting work is thought that this research can be a performance determination criterion in athletes. however, more research is needed.
A: Based on your suggestion and the suggestions of the other reviewers, we improved the introduction, including the literature review (61-91).
It is also necessary to integrate different tests that determine performance.
A: In our study, we conducted an observational match analysis of the world's best team handball players in the IHF World Championship. It is impossible to integrate different performance tests because it is not possible to measure these athletes, as they are playing in different clubs around the world and we have no chance to measure them. However, in a subsequent study in April 2023, we will compare the PlayerScore with different performance tests in an elite handball team. We will include a sentence about these perspectives in the discussion.

Reviewer 4 Report
Although the article is very interesting and suitable for any reader in this field, several recommendations should be considered for publication:
Relevant comments:
1. Evaluate the possibility of improving the abstract including the main results and conclusions of paper.
2. Introduction: improve the section include a literature review and the research objective.
3. The research method should be described in detail, thus allowing other researchers to replicate and build on published results.
4. Evaluate the possibility of including a methodology outline.
5. The section 2.4 Procedures does not add value.
6. The empirical research are the results must be improved. The quantitative results are poor, only take in account the PlayerScore.
7. Discussion: this section must to be improve, only compare PlayerSocre. One option could be a comparative study of the results obtained. The discussion must be a quantitative study.
8. Include ANOVA results comparative in the discussion.
9. Improve the scientific level of manuscript with more statistics measures.
10. The conclusions are basic. Improve this point. Include the highlights of the results obtained.
11. Few references to articles published in peer-reviewed journals have been included. Evaluate the possibility of including more references about this field, for example:
Martín-Guzón, I.; Muñoz, A.; Lorenzo-Calvo, J.; Muriarte, D.; Marquina, M.; de la Rubia, A. Injury Prevalence of the Lower Limbs in Handball Players: A Systematic Review. Int. J. Environ. Res. Public Health 2022, 19, 332. https://doi.org/10.3390/ijerph19010332
Bragazzi, N.L.; Rouissi, M.; Hermassi, S.; Chamari, K. Resistance Training and Handball Players’ Isokinetic, Isometric and Maximal Strength, Muscle Power and Throwing Ball Velocity: A Systematic Review and Meta-Analysis. Int. J. Environ. Res. Public Health 2020, 17, 2663. https://doi.org/10.3390/ijerph17082663
Minor comments:
1. Don`t write in the first-person point of view, for example “we conclude…” the correct way is with impersonal constructions.
2. there are many formatting errors: blank pages, incorrectly formatted table 2, incorrect spacing...
3. Figure 1b not appear.
4. Check references, there are mistakes. They do not adapt to the MDPI format. For example, the correct format of paper is: Author 1, A.B.; Author 2, C.D. Title of the article. Abbreviated Journal Name Year, Volume, page range.
Author Response
Reviewer #4:
Although the article is very interesting and suitable for any reader in this field, several recommendations should be considered for publication:
Relevant comments:
- Evaluate the possibility of improving the abstract including the main results and conclusions of paper.
A: We have included the main results in the abstract (lines 26-32) and a conclusion of the paper (lines 32-36); however, we used "conclude" instead of "suggest" to clarify that this sentence is the conclusion of the paper. Based on the comments of Reviewer #1, we reduced the abstract to 207 words!
- Introduction: improve the section include a literature review and the research objective.
A: Based on your suggestion and the suggestions of the other reviewers, we improved the introduction, including a literature review (L61-91) and the research objective (L118-126).
- The research method should be described in detail, thus allowing other researchers to replicate and build on published results.
A: Based on your suggestion and the suggestions of the other reviewers, we improved the methods, providing more information about the selection process and a better description to enable other researchers to replicate the study (L137-426).
- Evaluate the possibility of including a methodology outline.
A: As per reviewer´s suggestion we included a methodology outline. Thank you for this very helpful comment (Figure 2).
- The section 2.4 Procedures does not add value.
A: As per reviewer´s suggestion we rephrased the procedures, including the methodology outline (L386-406).
- The empirical research are the results must be improved. The quantitative results are poor, only take in account the PlayerScore.
A: The aim of the study was to validate the PlayerScore; consequently, we have only analyzed the PlayerScore. However, we have also added the detailed results of the best five players to the results (L450-455) in order to improve the results and discussion (L501-506) of the study.
- Discussion: this section must to be improve, only compare PlayerSocre. One option could be a comparative study of the results obtained. The discussion must be a quantitative study.
A: We have improved the discussion section based on your comments and those of the other reviewers (L468-586).
- Include ANOVA results comparative in the discussion.
A: We included the ANOVA results in the discussion (L540-542).
- Improve the scientific level of manuscript with more statistics measures.
A: It is not clear to us which other statistics we should include in the study, as we have calculated all relevant statistics; however, we have added some detailed information about the results from selected players (the top five ranked players) (L450-455).
- The conclusions are basic. Improve this point. Include the highlights of the results obtained.
A: As per your suggestion and that of Reviewer #1, we have rephrased the conclusion and added a sentence about the advantage of the PlayerScore compared to previous methods. However, the conclusion must be kept short and concise (comment of Reviewer #1).
- Few references to articles published in peer-reviewed journals have been included. Evaluate the possibility of including more references about this field, for example:
Martín-Guzón, I.; Muñoz, A.; Lorenzo-Calvo, J.; Muriarte, D.; Marquina, M.; de la Rubia, A. Injury Prevalence of the Lower Limbs in Handball Players: A Systematic Review. Int. J. Environ. Res. Public Health 2022, 19, 332. https://doi.org/10.3390/ijerph19010332
Bragazzi, N.L.; Rouissi, M.; Hermassi, S.; Chamari, K. Resistance Training and Handball Players’ Isokinetic, Isometric and Maximal Strength, Muscle Power and Throwing Ball Velocity: A Systematic Review and Meta-Analysis. Int. J. Environ. Res. Public Health 2020, 17, 2663. https://doi.org/10.3390/ijerph17082663
A: Based on your suggestion and the suggestions of the other reviewers, we included a total of 41 more relevant references (L613-703) about this field; however, your suggested reference (Martin-Gozon et al., 2022) deals with injury prevalence in team handball and is not particularly relevant in this context, so we have added the second reference (Bragazzi et al., 2020) to the discussion (L567-571)
Minor comments:
- Don`t write in the first-person point of view, for example “we conclude…” the correct way is with impersonal constructions.
A: It is not unusual to write in the first person; you will find a lot of papers using this spelling, and none of the other five reviewers criticized this spelling. We have written only a few sentences in the first person; hopefully, it will be okay for you if we do not change these sentences.
- there are many formatting errors: blank pages, incorrectly formatted table 2, incorrect spacing...
- Figure 1b not appear.
A: The format of the manuscript was changed after it was resubmitted by the editor of the journal; as a result, there were formatting errors. In the resubmitted manuscript, we adjusted the format and included all figures and tables.
- Check references, there are mistakes. They do not adapt to the MDPI format. For example, the correct format of paper is: Author 1, A.B.; Author 2, C.D. Title of the article. Abbreviated Journal Name Year, Volume, page range
A: We have adapted the reference style based on the citations style guide of the journal (L613-706).

Reviewer 5 Report
Thank you for the effort you put into the study titled ''The PlayerScore: A systematic game observation tool to determine the individual players performance in team handball competition ''.
How did you determine the number of participants in your research? Please inform the reader about the details in the required section.
Inform the reader about the inclusion, exclusion and exclusion criteria of the participants in your study.
It was mentioned that the research data were taken from the European and World Championships. Does the fact that the World Championships are of different level and quality from the European championships add a critical dimension to the research or not?
It is suggested that the limited aspects of the research be increased a little more.
It is also suggested that the discussion section be further developed with current articles.
Author Response
Reviewer #5:
Thank you for the effort you put into the study titled ''The PlayerScore: A systematic game observation tool to determine the individual players performance in team handball competition ''.
How did you determine the number of participants in your research? Please inform the reader about the details in the required section.
A: As per the reviewer's suggestion, we explained how we determined the number of participants (L160-166).
Inform the reader about the inclusion, exclusion and exclusion criteria of the participants in your study.
A: As per the reviewer's suggestion, we explained the inclusion and exclusion criteria of the participants in our study (L160-166).
It was mentioned that the research data were taken from the European and World Championships. Does the fact that the World Championships are of different level and quality from the European championships add a critical dimension to the research or not?
A: The best handball teams in the world are the European teams (e.g., in the current World Championship 2023, only the European teams Sweden, Denmark, Spain, and France are playing in the half-final), with only a few teams from Africa (Egypt, Tunisia), South America (Brazil, Argentina), and Asia (South Korea and Japan) playing on a similar level. Consequently, the European and World Championships are on a similar level, as the world's best teams (from Europe) are playing in both tournaments. Consequently, the answer to your question is no; the different tournaments do not add a critical dimension to the research.
It is suggested that the limited aspects of the research be increased a little more.
A: As per the reviewer's suggestion, we added more information about the limitations of the study (L565-586).
It is also suggested that the discussion section be further developed with current articles.
A: As per the reviewer's suggestion, we discussed our results in relation to previous studies. However, no previous study in team handball is comparable to this new analysis tool (PlayerScore) in team handball (L572-586).

Reviewer 6 Report
In my opinion the introduction is too basic with the information on the sport of handball. There is need for more theory why this research is so important. In addition, there are already pieces of information about the methods as well as the results in the first section of the paper. Overall the paper does not have an adequate structure for a publication. Therefore I suggest to reorder the information and give more details on the theory as well as the discussion. It is useful to know what are the benefits for the sport of handball, the education for coaches as well as players and what are further research areas that this paper offers.
I really like the table that you used in the method section. In my opinion you should use this and just describe it for this section.
Author Response
Reviewer #6:
In my opinion the introduction is too basic with the information on the sport of handball. There is need for more theory why this research is so important.
A: Based on your suggestion and the suggestions of the other reviewers, we improved the introduction, including a literature review and a better justification of the study (L61-91).
In addition, there are already pieces of information about the methods as well as the results in the first section of the paper.
A: As per the reviewer's suggestion, we reordered the introduction and the methods (L40-427). We are not sure what you mean by results in the first section of the paper? If you are referring to the results of the European Championship in 2020, it was important to show these results from a previous paper to explain why we have used the PlayerScore (L102-109).
Overall the paper does not have an adequate structure for a publication. Therefore I suggest to reorder the information and give more details on the theory as well as the discussion.
A: Based on your suggestion and the suggestions of the other reviewers, we reorganized the structure of the manuscript and provided more details on the theory as well as the discussion. Thank you for these helpful comments.
It is useful to know what are the benefits for the sport of handball, the education for coaches as well as players and what are further research areas that this paper offers.
A: You, as well as reviewers #1 and #4, are absolutely right; the discussion needs more information about the advantages of the PlayerScore compared to other match analysis methods. Consequently, we have added an additional paragraph to the resubmitted manuscript (L572-586).
I really like the table that you used in the method section. In my opinion you should use this and just describe it for this section.
A: Thank you for the compliment. I think you are referring to Table 2? We have described the table in this section, but it is also important to explain how we defined the variables and determined the scores.

Round 2
Reviewer 1 Report
Dear Authors,
It is noted that they have extensively revised the manuscript, so I have no further revisions to propose.
Well done!
Reviewer 4 Report
The authors have worked satisfactorily on this manuscript. In the opinion of this reviewer, the manuscript can be acceptted in present form.
Thanks
Reviewer 6 Report
Thank you for the changes. In my opinion this paper improved a lot and will be good to go for a publication.